# Effects of Prickly Burnet (*Sarcopoterium spinosum* (L.) Spach.) Control and Sheep Grazing on Hay Yield and Quality on Gökçeada Island, Turkey

**DOI:** 10.3390/ani12223073

**Published:** 2022-11-08

**Authors:** Fırat Alatürk, Ahmet Gökkuş, Altıngül Özaslan Parlak, Harun Baytekin, Cemil Tölü

**Affiliations:** 1Department of Field Crops, Faculty of Agriculture, Canakkale Onsekiz Mart University, Çanakkale 17100, Turkey; 2Department of Animal Science, Faculty of Agriculture, Canakkale Onsekiz Mart University, Çanakkale 17100, Turkey

**Keywords:** *Sarcopoterium spinosum*, pulling out, burning, cutting, botanical composition, hay yield, gökçeada island, sheep, grazing

## Abstract

**Simple Summary:**

This study was carried out to control the plant *Sarcopoterium spinosum* in Çanakkale, Turkey. As a result of grazing, the number of grasses and legumes decreased and the number of shrubs increased. The number of species increased due to shrub control measures but decreased remarkably with grazing.

**Abstract:**

(1) Background: The effects of prickly burnet (*Sarcopoterium spinosum*) control measures (pulling out, burning and cutting) and grazing on the botanical composition, grazeable dry matter (GDM) yield and nutritional values of rangeland were investigated on Imbros Island (Turkey) in 2010–2013. (2) Methods: The rangeland was grazed by Imbros sheep during the last year of the experiment. About 0.15 ha of rangeland was allocated to each sheep and five sheep were placed in each plot. Grazing was continuous throughout the year. (3) Results: Shrub levels decreased by 50–60% due to pulling out, burning and cutting in the first year and herbaceous species increased. Forbs increased more in the pulled and burnt plots and grasses increased more in the cut plots. In the third year, the shrub level increased to 60–65% and herbaceous species decreased. The decrease in herbaceous species was observed mostly in forbs. Plant cover was mostly (58%) composed of annual species. Development decreased plant cover ratios, but this decrease ceased in the burnt plot in the third year. Grazing also reduced plant cover. Crude protein (CP), NDF, ADF and digestible dry matter (DDM) content did not vary significantly over the experimental period. CP and DDM increased, NDF decreased and ADF did not change in the development plots. Overall, significant differences were not observed in GDM yield and nutritional values due to development efforts (pulling out, burning and cutting). (4) Conclusions: However, cutting is difficult over stony and rough terrain and pulling out creates erosion on sloping surfaces. Therefore, burning is recommended over the entire rangeland and burning or pulling out is recommended over smooth terrain for the temporary control of *S. spinosum*.

## 1. Introduction

Coastal western Turkey is an area with shrubbery plant cover composed of specific species based on climatic and soil factors which have developed in the Mediterranean, Aegean and Marmara regions. Imbros Island (formerly Imbros) is located in the Mediterranean climate zone and the majority of the island’s rangeland is covered with prickly burnet (*S. spinosum*) shrub [1]. The climate (hot and dry summers, and cool winters) and soil (shallow, infertile and limy) facilitated the extensive spread of the shrub over the island rangeland [2,3,4]. Since prickly burnet has high propagation due to its seeds and vegetative tissues [5,6], it can easily expand over rangeland. Prickly burnet forms compact thorns and hides its leaves among these thorns. It is also a distasteful species [2,7,8]. Therefore, animals have a hard time grazing on it and do not particularly want to graze on this plant. Given this resistance to grazing [9], the species has spread uncontrollably and reduced the availability of rangeland for grazing. Since the plants cover the soil surface substantially, they disturb the survival of other easily grazed herbaceous species. Such herbaceous species appear only in small spots and the tall species (generally perennials) grow upwards through the shrub to benefit from the sun. Research has been conducted, especially in Israel and Greece, to overcome the prickly burnet problem on rangeland. Various development methods, such as burning [2,10,11], treatment with herbicides [10,11] and mechanical methods using cutting or pulling out and fertilization [10] were applied either alone or in combination. Prickly burnet was not able to be permanently removed from the plant cover in these studies; it could only be suppressed for a certain period of time. Therefore, it was possible to benefit from the rangeland only during these periods. The objective of the present study was to improve vegetation cover by removing prickly burnet using pulling out, burning and cutting methods to allow the growth of local herbaceous species. Among herbaceous species in the Mediterranean zone plant cover, annual species constitute a significant proportion (about half) [7,8,12] and these species can rapidly germinate and cover large areas when the required conditions are met. The Imbros rangelands are open to free and continuous grazing and there are breeds of site-specific sheep and goats on the island [13]. These breeds are able to survive on hard-to-graze plant cover. Although they are not able to produce a high yield through grazing prickly burnet shrubs, they can benefit from the rangeland more than other breeds. Their small size also reduces their feed requirements [14,15]. For this reason, the existing Imbros breed of sheep was grazed over the development land in our study.

The present study is the first case study carried out on the development of Mediterranean shrubbery rangelands (garrigue). In this study, the effects of shrub controlling and grazing on the yield, species composition and nutritional values of the vegetation cover were investigated.

## 2. Materials and Methods

Place of study: This experiment has been conducted in the rangeland area of Yıldız Koyu (40°14′10.82″ N latitude, 25°54′30.45″ E longitude) District of Imbros (Imbros) island situated in the province of Çanakkale, Turkey. The experiment has been carried out in the shrubby rangelands dominated with the dwarf shrubs (garrigue) at Yıldız Koyu, located near the Aşağı Bademli village of Gökçeada (Imbros) district of Çanakkale province for 3 years, between 15 October 2010 and 7 December 2013 in Turkey. The average altitude of the research site was 53 m. The average monthly temperature of Imbros during the last 20 years (1982–2012) was 15.3 °C. The experimental period (except for 2011) experienced higher average temperatures than the long-term averages. The long-term average annual total precipitation was 722.1 mm and the experimental years (except for 2011, 640.8 mm) had higher precipitation than the long-term average. Throughout the experiment, droughts were generally observed between May and September. The experiment provided adequately for the feed and shelter needs of the animals. Paddock gates were left open to allow the animals to move freely over the rangeland.

Management of sheep in grazing: In the project carried out on the rangeland plots established in the upper part of the Yıldız Koyu (the name of the site where the experiment was established), located in the north of Gökçeada (Imbros) island, sheep (Gökçeada sheep) were selected from among a herd of 150 sheep roaming freely in rangeland, to come to the corral when “called (with the sounds of the sheep owner)” and were accustomed to being fed, and were randomly distributed to plots depending on their age, live weight and condition. A total of 40 randomly selected sheep were between the age of 3–4 years and had an average live weight of 31.18 kg. Each grazing plot was placed on an area with 7.5 ha and the protected plots consisted of 1.5 ha. Free grazing was conducted in the rangelands of Gökçeada (Imbros) Island. For this reason, the fencing was erected around the plots, primarily to protect the planted plants from animals in the experimental field. The process of fencing was conducted from 16–18 February 2011 by using a wicker iron cage with a length of 5 m and a height of 1.1 m. Paddocks and fences were established in the interior parts of the plots for grazing. They were placed just before grazing in order not to damage the natural form of rangeland. One side of the paddock was left open so that the animals would graze freely in the rangeland. A total of eight paddocks were placed, one for each application, in order to meet the physiological needs of the sheep. The paddock area was planned to be 1.5 m^2^ for each sheep by taking into account the birth and lactation period of the animals [16]. Accordingly, each paddock had an area of 7.5 m^2^. Approximately 2–3 year old pregnant Imbros sheep were used in the grazing treatments and about 0.15 ha of rangeland was allocated to each sheep and five sheep were placed in each plot measuring 0.75 ha. Paddocks were placed in each plot to meet the water supply needs of the sheep in December 2013 and the sheep roamed at will over the rangeland throughout this period.

Soil Sampling: In the experiment, which was established according to the randomized complete block design, improvement practices (burning, pulling, cutting and control) were placed on the main plots while the seeding and grazing practices were placed into the sub plots. Accordingly, the experiment trials consisted of a total of 16 plots which included 4 improvement methods*seedings*2 grazing. A total of 4 samplings were obtained from each plot and the plotting of the area was conducted from 23–24 October 2010. The first sampling was taken in October 2010, just after the plotting process, from the experiment area without conducting further practices. Soil sampling was conducted by taking into account the method of Jackson (1958). Just after the first sampling, the practices of burning, pulling, cutting and leaving natural were applied in the experiment plots. The second sampling was conducted in the following year on October 2011, and half of each experiment plot was seeded with forage crops just after taking the samples. The third sampling was conducted after one year in October 2012, and the sheep were released into half of each experiment plot and the grazing practice was conducted just after taking the samples. The fourth sampling was conducted one year after starting the grazing practice in November 2013. The analysis methods used while taking a total of 256 (4 years*64 samples) soil samples, over the period of 4 years, were carried out as stated below. The taken soil samples were brought to the soil laboratory of the Çanakkale Onsekiz Mart University, Faculty of Agriculture, Department of Soil Sciences and Plant Nutrition and awaited the air-drying process. The soil samples, after the air-drying process, were pounded and sieved through a 2 mm sieve and were prepared for further analysis. The experimental soils were slightly water-soluble alkaline (pH 7.92) with low water-soluble salts (94.46 ppm), low carbonate levels (2.32%), medium organic matter content (2.68%), very low P (2.42 ppm), very high Ca (42.015 ppm), Mg (4275 ppm) and K (2823 ppm) content and low Na (675 ppm).

Composition of Plant Species: The distribution of species constituting the vegetation was determined in the experiment. Vegetation cover was measured using the ring (loop) method developed by Parker and Harris [17]. In each plot, 10 tracks were measured. The measuring tracks were fixed and the same tracks were measured throughout the experiment.

Hay parameter: In each year, at the beginning of March, at the end of May, at the beginning of October and at the beginning of December, in each sub plot 5 m^2^ areas from each were cut from the bottom four times in a year. Moreover, the protected area, created to protect the plants from grazing, was cut in order to accurately measure the yield in the grazed plots. The cutting practice was performed manually, using a motorized portable hand harvester and with the help of cutting scissors. Harvested hay was dried in the open air first, and then they were placed in an oven set at 65 °C [18]. Afterwards, the hay yield of the rangeland in kg/ha was calculated by weighing the hay, and the average was taken. Plant samplings were conducted in the years 2012 and 2013 in order to determine the hay yields. In the first year of grazing, the samplings were conducted twice in the months of May and November while in the last year of grazing, the samplings were conducted thrice in the months of March, May and November in 2013. Plant samplings were conducted from both protected as well as grazing plots. Firstly, the taken hay samples were dried and then weighed. Lastly, the amount of grazing was calculated using the difference in yield between the protected and grazed areas as indicated in the formula given below.
Amount of consumed hay (kg/ha) = Yield of protected area (kg/ha) − Yield of grazed area (kg/ha)

Applications: Four development methods (pulling out, burning, cutting and natural (control)) were used to combat the prickly burnet shrub in this study. For the pulling out method, the soil was tilled at 10–15 cm depth with a chisel plow. At the end of this process, about 70% of the prickly burnet shrubs had been removed. Pulled shrubs were removed from the field. A double-chain shrub cutter was used for the cutting. Since the rangeland has a stony surface, shrubs were cut 10–15 cm above the ground. Parts of the prickly burnet shrub in the surface soil were burnt easily and rapidly while taking the required safety precautions. These development processes were practiced only at the beginning of the experiment (23 October 2010). The botanical composition, species diversity, soil cover ratio and grazeable dry matter yield of the vegetation were also investigated in this study. Botanical composition was measured by the loop method development by [17]. In each plot, 10-line measurements were performed in May 2011–2013. The plant cover ratio was calculated as the ratio of measurement points in which plants were encountered to the total measurement points. For grazeable dry matter yield, 5 sections of 1 m^2^ were mowed from the bottom including prickly burnet shrubs. The mowed herbage was initially air-dried, then dried in an oven at 65 °C. Then, the herbage was weighed and the average was taken to calculate the dry matter yield in kg/ha. In these calculations, all herbaceous species and 1/10 of prickly burnet were considered grazeable. Dry-matter yields were determined in 2012 and 2013. Plant samples were harvested from the rangeland and ground. Then, the crude protein ratios of the samples were determined by the Kjeldahl method [19], NDF and ADF analyses were carried out in accordance with the method recommended by [20], and digestible dry matter was determined using the ANKOM method (DaisyII-200/220 Incubator Operator’s Manual).

Statistical analysis: Experiment trials were carried out and established according to the Randomized Complete Block Design using five replications. During the evaluation of obtained data, depending on the purpose, the factorial analysis of variance technique (e.g., dry hay yield) [21], multidimensional scaling (MDS) (e.g., species distribution in vegetation cover) [22], the Z Test, χ2 analysis [21] and suitability analysis (correspondence) technique (rate of emergence of cultivated species) [23] were used in the experimental arrangement of random plots.

## 3. Results

### 3.1. Botanical Composition (%)

Most of the rangeland plant cover on Imbros was composed of short shrubs. Of these shrubs, more than 95% consisted of prickly burnet. While the shrubs constituted about 39.91–48.10% of the plants in the first year, after the pulling out, burning and cutting had taken place the ratio of shrubs increased to 59.42–67.20% in 2013. Grazing, on the other hand, reduced the herbaceous species and consequently increased the ratio of shrubs. Over the pulled, burnt and cut plots, the ratio of shrubs even reached 88.24–90.02% due to grazing. In the natural rangeland, almost all the plant cover was composed of shrub (prickly burnet) over the three years (Table 1). Thus, a grazing-induced change was not observed in the natural vegetation. Shrubs in the plant cover were generally represented by 2–3 species (*S. spinosum*, *Centaurea spinosa* var. Spinosa and *Thymus zygioides*). On the other hand, a total of 143 species were identified in the rangeland area throughout the experiment (Appendix A, Appendix B, Appendix C and Appendix D). Among the herbaceous species, grasses formed a significant percentage of the plant cover in the development plots. As an annual average, the ratio of grasses varied between 13.45% and 20.26% depending on development treatment. While the ratio of grasses increased in the pulled plots over the three years, the ratio exhibited a decreasing trend over the burnt and cut plots. Grazing significantly decreased the grass ratio in each of the three development methods. With grazing, the ratio of grasses decreased from 23.55% to 4.16% in the pulled plots, from 12.59% to 0.88% in the burnt plots, and from 20.72% to 5.82% in the cut plots. In the natural rangeland, on the other hand, herbaceous species including grasses were not able to survive (Table 1). Although varying according to the type of development used, mostly *Dactylis glomerata*, *Brachypodium distichum* and *Bromus tectorum* were encountered among the grass species. Removal of shrubs from the plant cover resulted in a significant increase in the number of grasses, but the number then decreased with grazing. Depending on the year and development practice, the number of grasses varied between 9 and 15 species (63.6–73.7% of which were annual species) and the number of grass species in the grazing plots varied between 4 and 6 (Appendix A, Appendix B and Appendix C). Legumes are noted for their delectability, high protein content and soil nitrogen enhancement and they were encountered at quite low levels in the plant cover. Contrary to forbs, legumes exhibited an increasing trend over the years in each of the three development treatments. The ratio of legumes increased from 0.90% to 6.52% as a result of pulling out, from 1.69% to 17.35% as a consequence of burning, and from 2.24% to 7.52% due to cutting. However, grazing also remarkably reduced this herbaceous species group. Following grazing, while the ratio of legumes in the plant cover of the pulled and burnt plots was observed to be 0.28% and 3.65%, respectively, legumes were totally removed from the cut rangeland by grazing. Legume species were not encountered in the natural rangeland. Among the legumes, *Ononis spinosa* was the most common species in the development plots and some Trifolium and Medicago species were also encountered in these plots. Based on the year and development method, the number of legume species varied between 3 and 11, of which about 54.5–69.2% were annual species (Appendix A, Appendix B and Appendix C). Except for natural rangeland, the ratio of forbs in the vegetation cover of the development plots varied from 19.96% to 46.68% at the beginning of the experiment and the values significantly decreased (to 4.56–10.51%) at the end of the experiment. The ratio of forbs did not change significantly with grazing. The proportion of forbs in the grazed plots varied between 5.54% and 5.94%. These species were not encountered in the natural rangeland in 2012. Within the vegetation cover of the rangeland plots, forbs had the greatest number of species. Based on year, the number of species varied between 12% and 29% (61.4–64.6% of which were annual/biennial). The species were represented by 5–7 species in the grazed plots. Only one species was encountered in the natural rangeland in 2012. In this group of plant species, *Asphodelus microcarpus*, *Anagallis caerulae*, *Anagallis arvensis* and *Crepis sancta* were the most common (Table 1; Appendix A, Appendix B and Appendix C). Annual/biennial species constituted the majority of the plant cover in the development plots (Figure 1). Of all plant species encountered in the experimental plots, 58% were annual and biennial species and 42% consisted of perennial species (Appendix A, Appendix B, Appendix C and Appendix D).

The total number of species in the plant cover of the development plots differed significantly according to development method. Such a difference resulted from the natural rangeland. The natural rangeland was represented generally by 2–3 species. On the other hand, the number of species increased in the development plots. However, distinctive differences were not observed in the number of species resulting from different development practices. However, significant differences were observed by year. While the number of species was between 39 and 45 in the first year, the figure increased to 55–58 in the second year. The number of species decreased again in the last year and was observed to be 33–34. A remarkable decrease was observed in the number of species due to grazing. Only about 13–15 species were encountered in the grazed plots (Figure 1).

### 3.2. Plant Cover Ratio (%)

Significant differences were observed in plant cover ratios according to year and development practice (*p* < 0.01). The greatest cover ratio throughout the experiment was observed in natural rangeland (90.0%, 93.5% and 93.0% in 2011, 2012 and 2013, respectively). Soil cover ratios in the pulled and burnt plots increased from the beginning of the experiment. In the pulled plots, a soil cover ratio of 66.4% in 2011 increased to 82.3% in 2013 and the figure for the burnt plots increased from 70.9% to 90.5%. The situation was the reverse in cut plots. The plant cover ratio of 89.2% in the first year decreased to 81.1% in the last year. On the other hand, a significant decrease was observed in plant cover ratios of the grazed plots. The soil cover ratios in the grazed natural, pulled, burnt and cut plots were observed to be 81.5%, 72.2%, 68.4% and 72.3%, respectively (Table 2).

### 3.3. Grazeable Dry Matter Yield (GDMY)

Since the shrubs in the experimental fields were short and thin, they were cut easily. Therefore, shrubs were also included in the dry matter yield of the rangeland, but only the grazeable sections of the shrubs were taken into consideration. The grazeable dry matter yield of the rangeland significantly varied by year (*p* < 0.01). The yield of 725.6 kg/ha in 2012 increased to 1001.0 kg/ha in 2013. The yield significantly decreased in the grazed plots. As for the annual average, the cut and burnt plots had the greatest dry matter yield with 954.7 and 919.0 kg/ha, respectively. In the first sampling year (2012), pulled plots produced significantly lower grazeable herbage than the other plots (*p* < 0.01). In 2013, the dry matter yield of the pulled, burnt and cut plots without grazing was higher than the natural rangeland. On the other hand, the situation reversed in the grazed plots (Table 3).

### 3.4. Nutritional Values (%)

Plant samples were taken to determine crude protein (CP), NDF, ADF and digestible dry matter (DDM). The crude protein ratios of samples mowed from the rangeland were not significantly different between 2012 (4.74%) and 2013 (4.67%) but significant differences were observed in the development areas (*p* < 0.01). While the crude protein ratio of the samples taken from development plots varied between 4.85% and 5.38%, the value was observed as 3.53% in samples taken from natural rangeland. The CP ratio of samples slightly increased with grazing (4.87%), but such an increase was not found to be significant. While the differences in NDF ratios by year and development practice were found to be significant (*p* < 0.01), the differences in ADF ratios were not significant. The NDF ratio was higher in the first year (58.88%) and lower in the second year (53.08%). The NDF ratio of grazed plots (with an average value of 54.29%) was in between the values of the two years. ADF ratios of grazed plots varied between 40.56% and 41.96%. On the other hand, NDF ratios of developments plots (53.64–54.93%) were lower than the NDF ratio of natural rangeland (58.60%). The ADF ratio of developments plots varied between 39.84% and 41.73%. DDM content was higher in the pulled, burnt and cut plots (46.04–49.18%) than in natural rangeland (43.07%). Significant differences were not observed in DDM content according to year or grazing (Table 4).

## 4. Discussion

Generally, shrubs constituted the largest plant group in all years and development treatments. Because they adapt well to the Mediterranean ecology and competitive conditions [2,3,4,6], prickly burnet exhibited a regular increase over the three years. On the other hand, uncompetitive herbaceous species decreased throughout the experimental period.

Depending on the development method, the ratio of shrubs in the rangeland plant cover decreased more than 50% in the first year and the ratio of herbaceous species increased in that year. In the later years, shrubs (especially prickly burnet) increased in all development plots and constituted about two-thirds of rangeland vegetation. Prickly burnet, with its highly adaptive capacity and competitive advantage in vegetation cover, temporarily decreased then increased again later on. It has been reported in similar studies that the effect of burning lasts about five years and that plants then return to their original (initial) ratio at the end of this period [11]. Since no method was practiced in the natural rangeland (control) to check the spread of this shrub on Imbros, a change was not observed in *S. spinosum* and *C. spinosa* cover in the natural rangeland.

Forbs were represented at their highest level in the rangeland vegetation during the first year of the experiment. Grasses took second place and legumes had the lowest ratio. As is normal in the Mediterranean climate zone, annual species that survive drought are common in ecologies where drought is frequent [24,25]. Since about 50% [12], 67% [7] or 74% [26] of species in the Mediterranean vegetation cover are said to be composed of annual species, the plant cover in these ecologies replenishes itself rapidly after fires, soil disinterment and grazing [27]. Since forbs constitute a large proportion of annual species, a rapid increase was observed in these species after shrub control measures. However, the increased prickly burnet cover in subsequent years again reduced the share of forbs in the vegetation cover. The change in grass species over time varied according to the development practice. While grasses continuously increased in the pulled plots, they decreased in the cut plots. In the burnt plots, they initially increased and then later decreased. Such variations in grasses mostly resulted from changes observed in the species *Brachypodium distichum*, *Catapodium rigidum*, *Aegilops* sp., *Avena barbata* and *Dactylis glomerata*.

Legumes continuously increased in all development plots. Since the legume *Ononis spinosa* has deep and strong roots, it was able to sustain itself within the prickly burnet shrubs. Therefore, an increase in legume ratio was observed in the pulled and burnt plots following shrub control measures. The increase in legume ratio in the cut plots mainly resulted from an increase in the annual Trifolium species. Annual species played a significant role in the change of herbaceous species, depending on the development practice. Climate is the greatest determinant in the emergence of these species. Fluctuations in climate significantly affect plant cover density, phenological growth stages, yield and propagation [21,26]. Climate was also found to be effective in the botanic composition of the present study.

As in all rangeland, grazing was the primary factor designating the vegetation model in Mediterranean rangelands [25]. Since shrubs have stiff structures that are resistant to harsh conditions, their proportion in the botanical composition continuously increased due to grazing; however, the ratio of herbaceous species decreased with grazing, confirming studies in other Mediterranean environments where the presence of wild grazers predominates over livestock [28,29,30]. Legumes, for instance, largely withdrew from the rangeland. The sheep preferred to graze on the herbaceous species, especially the legumes, and consumed almost all of them in one month (April).

Species diversity increased significantly due to the pulling out, burning and cutting measures implemented over the rangeland to control the prickly burnet. Removal of shrubs from the plant cover opened a survival area for herbaceous species. Because of the species in soil seed stocks and those species transported from elsewhere, the number of species rapidly increased. Such an increase turned into a decrease in 2013. In other words, the increase in the number of species resulting from development lasted only two years. However, the number of species at the end of three years was not reduced to the level in the natural rangeland. In one study, according to the development method, climate and soil conditions, it was observed that prickly burnet returned to its initial levels within 5–17 years [31]. Despite the decrease over time, species diversity in the development plots was more than the natural rangeland during this period.

Grazing decreased the number of species in all plots. The increase in species mainly resulted from herbaceous species. Since herbaceous species were intensely grazed by the sheep, these plants were not able to regenerate. Most of these were annual species. Heavy grazing both hinders seed development and seedling growth of these species [26]. Therefore, a decrease was observed in the species diversity of the rangeland [32].

Herbaceous species cannot compete with deep-rooted tall shrubs [33,34]. Therefore, ligneous species replace herbaceous species progressively throughout the vegetation cover process [20]. Since the plant cover changed according to the ratio of prickly burnet in the vegetation cover [35], the plant cover ratio increased concurrently with the increasing prickly burnet ratio. Thus, the plant cover ratio was lowest in the pulled and burnt plots with the least prickly burnet mass and highest in natural rangeland which had not undergone any form of development. In the grazing plots, the sheep consumed the easily grazed and digestible herbaceous species in a short time (about 1 month), then grazed on some prickly burnet. Thus, plant cover ratios decreased in the grazing plots.

Protection measures can improve the species diversity, dry matter production and plant density of rangelands [36,37,38]. In our study, the progressive increase in dry matter yield was due to the easy propagation of plants without their being under grazing pressure. Since herbaceous species grow faster than shrubs, such an increase was distinctive in the pulled, burnt and cut plots. Under heavy grazing conditions, plants may lose their photosynthetic tissues and reserve nutrients [32], thus dry matter production, species diversity and cover ratios usually decrease under these conditions [39]. Therefore, dry matter production significantly decreased in the grazed plots during the last year of the experiment. Since sheep are able to graze on herbaceous species more easily [40,41], a heavy grazing-induced yield decrease occurred more in the pulled, burnt and cut plots which had a high herbaceous species ratio.

Nutrient content, species composition, leaf-shoot ratio and plant growth stages usually vary based on soil and climatic conditions [27,42]. Herbaceous species with their soft tissues generally have higher protoplasm and less membrane components. Such a case results in higher CP and DDM content in herbaceous species during their green periods but lesser ADF and NDF ratios than several shrubs [8]. Since the ratio of herbaceous species increased in the development plots (Table 1), CP and DDM content of the plant samples taken from these plots was higher. On the other hand, lignification in the cell walls of ligneous species [43] is the greatest factor in decreasing their DDM content. Grazing did not significantly affect the nutritional values of the herbage. Since heavy grazing did not allow the regeneration of herbaceous species, grazing was not able to affect the nutrient content of the plants. It is recommended that the CP ratio of roughage should be at least 7% [44] and DDM content should be between 55% and 60% [45]. Although nutritional values of the herbage increased in the development plots of the present study, CP and DDM values were below the recommended values. Therefore, supplementary feed should be provided to animals grazing over these rangelands in every season of the year.

## 5. Conclusions

Development methods of pulling out, burning and cutting on rangeland with dominant prickly burnet cover were temporarily effective in controlling the shrub. Following development, the ratio of shrubs to plant cover continuously increased. Three-year botanical composition measurements revealed that the prickly burnet will cover the vegetation again in five years. Therefore, shrub control measures should be repeated at 5-year intervals to continuously and sufficiently benefit the rangeland. Shrub control rapidly increased species diversity within two years, and annual species were dominant in this increase. Grazeable dry matter yield increased because of the development but such an increase was insufficient because of the infertile ecosystem (soil). Therefore, in the case of continuous grazing, even in the most fertile year (2012), 0.65–0.7 ha of rangeland area was required per Imbros sheep and 8–8.5 ha rangeland area was necessary per animal. The herbage quality of the rangeland improved with the reduction of shrub in the plant cover. However, cutting is difficult over stony and rough terrain and pulling out creates erosion on sloping surfaces while burning the herbage quality was still not above recommended threshold values. Therefore, pasture quality for productive livestock may be insufficient in periods of the year.

## Figures and Tables

**Figure 1 animals-12-03073-f001:**
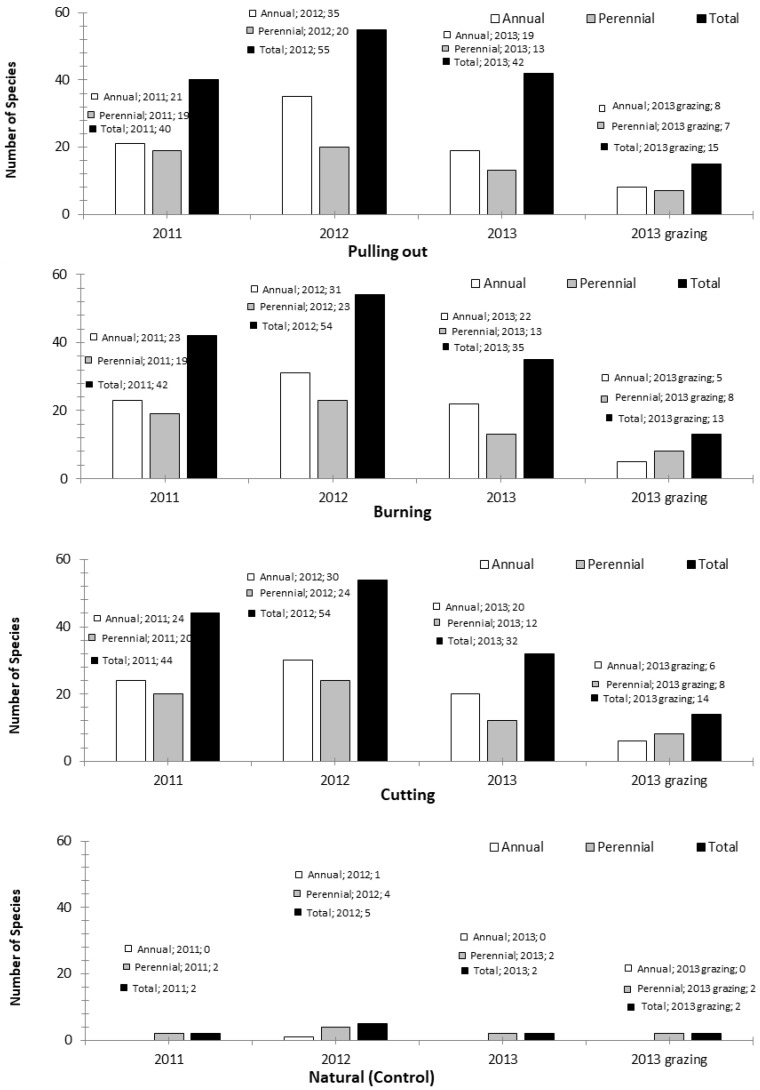
Number of annual/biennial and perennial species in vegetation cover according to development method.

**Table 1 animals-12-03073-t001:** Distribution of plant groups at improved and grazed rangeland levels according to year (% $-\bar{x}$ ± SD).

Plant Groups	2011	2012	2013	2013 (Grazing)	Mean
Grasses					
Pulling out	12.51 ± 2.48	13.61 ± 0.95	23.55 ± 6.60	4.16 ± 0.38	13.45 ^b^
Burning	19.18 ± 2.24	23.02 ± 5.71	12.59 ± 1.15	0.88 ± 1.94	13.92 ^b^
Cutting	32.40 ± 11.58	22.08 ± 5.04	20.72 ± 4.60	5.82 ± 1.56	20.26 ^a^
Natural	0.00 ± 11.33	1.07 ± 9.82	0.00 ± 10.05	0.00 ± 2.56	0.27 ^c^
Mean	16.02 ^a^	14.95 ^ab^	14.21 ^b^	3.62 ^c^	
Legumes					
Pulling out	0.90 ± 0.22	5.81 ± 1.03	6.52 ± 0.94	0.28 ± 0.49	3.38 ^b^
Burning	1.69 ± 0.34	7.16 ± 1.98	17.35 ± 6.72	3.65 ± 1.89	7.46 ^a^
Cutting	2.24 ± 0.73	4.47 ± 0.08	7.52 ± 0.23	0.00 ± 0.69	3.55 ^b^
Natural	0.00 ± 0.86	0.00 ± 3.08	0.00 ± 5.55	0.00 ± 0.69	0.00 ^c^
Mean	1.21 ^c^	4.36 ^b^	7.85 ^a^	0.98 ^c^	
Forbs					
Pulling out	46.68 ± 15.74	14.49 ± 1.72	10.51 ± 3.18	5.54 ± 0.86	19.31 ^a^
Burning	31.03 ± 4.67	18.98 ± 4.89	8.95 ± 2.08	5.85 ± 1.07	16.20 ^a^
Cutting	19.96 ± 3.15	11.03 ± 0.73	4.56 ± 1.03	5.94 ± 1.14	10.37 ^b^
Natural	0.00 ± 17.27	3.74 ± 5.88	0.00 ± 4.25	0.00 ± 3.06	0.94 ^c^
Mean	24.42 ^a^	12.06 ^b^	6.01 ^c^	4.33 ^c^	
Shrubs					
Pulling out	39.91 ± 13.04	66.09 ± 1.80	59.42 ± 8.85	90.02 ± 1.37	63.86 ^bc^
Burning	48.10 ± 7.25	50.84 ± 12.58	61.11 ± 7.65	89.62 ± 1.65	62.42 ^c^
Cutting	45.40 ± 9.16	62.42 ± 4.39	67.20 ± 3.34	88.24 ± 2.63	65.82 ^b^
Natural	100.00 ± 29.45	95.19 ± 18.78	100.00 ± 19.85	100.00 ± 5.69	98.79 ^a^
Mean	58.35 ^d^	68.63 ^c^	71.93 ^b^	91.96 ^a^	

^a–d^: Different means are indicated with different letters (*p* < 0.05).

**Table 2 animals-12-03073-t002:** Plant cover of improved and grazed rangeland (%).

	2011	2012	2013	2013 (Grazing)	Mean
Pulling out	66.4	80.8	82.8	72.2	75.5 ^c^
Burning	70.9	77.4	90.5	68.4	76.8 ^c^
Cutting	89.2	80.6	81.1	72.3	81.0 ^b^
Natural	90.0	93.5	93.0	81.5	89.5 ^a^
Mean	79.1 ^b^	83.1 ^b^	86.9 ^a^	73.6 ^c^	

^a–c^: Different means are indicated with different letters (*p* < 0.05).

**Table 3 animals-12-03073-t003:** Grazeable dry matter yield of improved and grazed rangeland (kg/ha).

	2012	2013	2013 (Grazing)	Mean
Pulling out	556.7	1149.1	425.2	852.9 ^ab^
Burning	798.4	1039.6	423.6	919.0 ^a^
Cutting	822.6	1086.8	543.5	954.7 ^a^
Natural	724.6	728.3	775.0	726.5 ^b^
Mean	725.6 ^b^	1001.0 ^a^	541.8 ^c^	890.5

^a–c^: Different means are indicated with different letters (*p* < 0.05).

**Table 4 animals-12-03073-t004:** Amount of CP, NDF, ADF and DDM of herbage harvested from improved and grazed rangeland.

	2012	2013	2013 (Grazing)	Mean
Crude protein (%)				
Pulling out	5.46	5.06	5.62	5.38 ^a^
Burning	5.16	5.26	5.42	5.28 ^a^
Cutting	4.87	4.77	4.92	4.85 ^a^
Natural	3.48	3.61	3.52	3.53 ^b^
Mean	4.74	4.67	4.87	
NDF (%)				
Pulling out	57.93	52.52	53.10	54.52 ^b^
Burning	56.81	52.08	52.03	53.64 ^b^
Cutting	59.35	52.27	53.16	54.93 ^b^
Natural	61.44	55.47	58.89	58.60 ^a^
Mean	58.88 ^a^	53.08 ^b^	54.29 ^ab^	
ADF (%)				
Pulling out	41.15	40.88	40.71	40.91
Burning	38.86	39.80	40.86	39.84
Cutting	41.70	40.80	42.68	41.73
Natural	40.51	41.01	43.61	41.71
Mean	40.56	40.62	41.96	
Digestible dry matter (%)				
Pulling out	48.29	48.82	45.39	47.54 ^a^
Burning	48.24	48.00	51.23	49.18 ^a^
Cutting	41.83	52.20	44.36	46.04 ^ab^
Natural	42.15	42.79	44.36	43.07 ^b^
Mean	45.13	47.95	46.33	

^a,b^: Different means are indicated with different letters (*p* < 0.05).

## Data Availability

Not applicable.

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
