# Peer review of "Effects of Prickly Burnet (Sarcopoterium spinosum (L.) Spach.) Control and Sheep Grazing on Hay Yield and Quality on Gökçeada Island, Turkey"

_animals, 2022, doi:10.3390/ani12223073_

Round 1
Reviewer 1 Report
It is necessary to make a description of the sheep, including the breed, age, body weight, productive management. The welfare standard and ethical norms that were followed during the development of the experiment should be included. There is no order in the methodology, it is necessary to organize all the information that was made. The procedure can be done with subchapters, eg.
Standard of ethical management of animals. B) Place of study. C) Management of sheep in grazing, D) Sampling techniques in forage and soil, E) Laboratory analysis. F) Statistical analysis. A statistical model must be included.
Include details of plant collections, conservation techniques, and identifications
Results. The authors must follow the same order of the methodology.
Figure 1 should not be repeated twice in the sampling year.
The figures do not have footnotes where the meaning of the letters is indicated and to what degree of significance they were compared.
It is necessary for the authors to determine the total digestible nutrients and amounts of total and digestible energy of the groups of plants. They can do it with formulas.
The authors conclude the use of a concentrate supplement for sheep. However, they do not make a deep discussion of the contribution of energy and consumption of plants. It is necessary to make an estimate of the requirements of the sheep.
Author Response
Response to Reviewer-1
1.It is necessary to make a description of the sheep, including the breed, age, body weight, productive management. The welfare standard and ethical norms that were followed during the development of the experiment should be included. There is no order in the methodology, it is necessary to organize all the information that was made. The procedure can be done with subchapters, eg.
---- Sheep (Gökçeada sheep) in the project carried out on the rangeland plots established in the upper part of the Yıldız Koyu (the name of the site where experiment was established), located in the north of Gökçeada (Imbroz) island, were selected from among a herd of 150 sheep roaming freely in rangeland, come to the corral when “called (with the sounds of the sheep owner)” and are accustomed to being fed, and were randomly distributed to plots depending on their age, live weight and condition. A total of 40 randomly selected sheep were between the age of 3-4 years and has an average live weight of 31.18 kg. Each grazing plot is placed on an area with 7.5 da and the protected plots were consisted of 1.5 da. Free grazing is done in the rangelands of Gökçeada (Imbroz) island. For this reason, the fencing was done around the plots, primarily to protect the planted plants from animals in the experimental field. The process of fencing has been done between 16-18.02.2011 by using a wicker iron cage with a length of 5 m and a height of 1.1 m. Paddocks and fences were established into the interior parts of the plots for grazing were placed just before grazing in order not to damage the natural form of rangeland. One side of the paddocks has left open so that the animals would graze freely in the rangeland. That is why, a total of eight paddocks were placed, one for each application, in order to meet the physiological needs of the sheep. The paddock area is planned to be 1.5 m2 for each sheep by taking into account the birth and lactation period of the animals (Altan et al., 2006).
2.Standard of ethical management of animals. B) Place of study. C) Management of sheep in grazing, D) Sampling techniques in forage and soil, E) Laboratory analysis. F) Statistical analysis. A statistical model must be included.
- B) Place of study: The experiment has been carried out in the shrubby rangelands dominated with the dwarf shrubs (garig) at Yıldız Koyu, located near the AÅŸağı Bademli village of Gökçeada (Imbroz) district of Çanakkale province for 3 years, between 15 October 2010 - 07 December 2013 in Turkiye.
- C) Management of sheep in grazing: Free grazing is done in the rangelands of Gökçeada (Imbroz) island. For this reason, the fencing was done around the plots, primarily to protect the planted plants from animals in the experimental field. The process of fencing has been done between 16-18.02.2011 by using a wicker iron cage with a length of 5 m and a height of 1.1 m. Paddocks and fences were established into the interior parts of the plots for grazing were placed just before grazing in order not to damage the natural form of rangeland. One side of the paddocks has left open so that the animals would graze freely in the rangeland. That is why, a total of eight paddocks were placed, one for each application, in order to meet the physiological needs of the sheep. The paddock area is planned to be 1.5 m2 for each sheep by taking into account the birth and lactation period of the animals (Altan et al., 2006).
- D) Sampling techniques in forage and soil:
Soil Sampling: In the experiment, which was established according to the randomized complete block design, improvement practices (burning, pulling, cutting and control) were placed on the main plots while the seeding and grazing were into the sub plots. Accordingly, the experiment trials were consisted of a total of 16 plots which included with 4 improvement methods x 2 seedings x 2 grazing. 4 samplings were done from each plot and the plotting of the area was done on 23-24 October 2010. First sampling was taken in October 2010, just after the plotting process, from the experiment area without doing further practices. Soil sampling was done by taking into account the method of Jackson (1958). Just after the first sampling, the practices of burning, pulling, cutting and leaving natural were applied in the experiment plots. Second sampling was done in the following year on October 2011, and half of each experiment plot was seeded with forage crops just after taking the samples. Third sampling was done after one year on October 2012, and the sheep were released into half of each experiment plot and the grazing practice was done just after taking the samples. Forth sampling was done after one year after started the grazing practice on November 2013. The analyzes and analysis methods used while taking a total of 256 (4 years x 64 samples) soil samples, over the period of 4 years, were carried out as stated below. The taken soil samples were brought to the soil laboratory of the Canakkale Onsekiz Mart University, Faculty of Agriculture, Department of Soil Sciences and Plant Nutrition and waited for air dry process. Soil samples, after air dry process, were pounded and sieved through a 2 mm sieve and got ready them for further analysis.
Composition of Plant Species: Distribution of species constituting the vegetation was determined in the experiment. Vegetation cover was measured using the ring (loop) method developed by Parker and Harris (1959) and recommended by Bakır (1970) for arid rangelands in our country, and used by many researchers (GökkuÅŸ, 1994; Şılbır and Polat, 1996; Koç and Çakal, 2004) in their researches. In each plot, 10 tracks were measured. The measuring tracks were fixed and the same tracks were measured throughout the experiment.
Hay Yield: In each year, at the beginning of March, at the end of May, at the beginning of October and at the beginning of December in each sub plot 5 m2 areas from each were cut from the bottom four times in a year. Moreover, the protected area, created to protect the plants from grazing, was cut in order to accurately measure the yield into the grazed plots. Cutting practice was done manually, a motorized portable hand harvester and with the help of cutting scissors. Harvested hay were dried in open air first, and then they were placed in an oven set at 65 °C (Cook and Stubbendieck, 1986). Afterwards, the hay yield of the rangeland in kg/da was calculated by weighing them, and then took the average.
Amount of Consumed Hay: Plant samplings were done in the years 2012 and 2013 in order to determine the hay yields. In the first year of grazing, the samplings were done twice in the months of May and November while in the last year of grazing, the samplings were done thrice in the months of March, May and November in 2013. Plant samplings were done from both protected as well as grazing plots. Firstly, the taken hay samples were dried and then weighed. Lastly, the amount of grazing was calculated from the difference in yield between the protected and grazed areas as indicated in the formula given below.
Amount of consumed hay (kg/da) = Yield of protected area (kg/da) – Yield of grazed area (kg/da)
- F) Statistical analysis. A statistical model must be included: Experiment trials were carried out and established according to the Randomized Complete Block Design using five replications. During the evaluation of obtained data, depending on the purpose, the factorial analysis of variance technique (e.g., dry hay yield) (Zar, 1999), multidimensional scaling (MDS) (e.g., species distribution in vegetation cover) (Kruskal, 1964; BaÅŸpınar et al., 2000), Z Test, χ2 analysis (Zar, 1999) and suitability analysis (correspondence) technique (rate of emergence of cultivated species) (BaÅŸpınar and MendeÅŸ, 2000) were used in the experimental arrangement of random plots.
3.Include details of plant collections, conservation techniques, and identifications
Composition of Plant Species: Distribution of species constituting the vegetation was determined in the experiment. Vegetation cover was measured using the ring (loop) method developed by Parker and Harris (1959) and recommended by Bakır (1970) for arid rangelands in our country, and used by many researchers (GökkuÅŸ, 1994; Şılbır and Polat, 1996; Koç and Çakal, 2004) in their researches. In each plot, 10 tracks were measured. The measuring tracks were fixed and the same tracks were measured throughout the experiment.
Hay Yield: In each year, at the beginning of March, at the end of May, at the beginning of October and at the beginning of December in each sub plot 5 m2 areas from each were cut from the bottom four times in a year. Moreover, the protected area, created to protect the plants from grazing, was cut in order to accurately measure the yield into the grazed plots. Cutting practice was done manually, a motorized portable hand harvester and with the help of cutting scissors. Harvested hay were dried in open air first, and then they were placed in an oven set at 65 °C (Cook and Stubbendieck, 1986). Afterwards, the hay yield of the rangeland in kg/da was calculated by weighing them, and then took the average.
Amount of Consumed Hay: Plant samplings were done in the years 2012 and 2013 in order to determine the hay yields. In the first year of grazing, the samplings were done twice in the months of May and November while in the last year of grazing, the samplings were done thrice in the months of March, May and November in 2013. Plant samplings were done from both protected as well as grazing plots. Firstly, the taken hay samples were dried and then weighed. Lastly, the amount of grazing was calculated from the difference in yield between the protected and grazed areas as indicated in the formula given below.
Amount of consumed hay (kg/da) = Yield of protected area (kg/da) – Yield of grazed area (kg/da)
- Results. The authors must follow the same order of the methodology.
---Necessary corrections are included into the text of the manuscript.
- Figure 1 should not be repeated twice in the sampling year.
----In this part of the study, the grazing practice was done in 2013 and the changes in the vegetation cover, due to grazing, were investigated. That is why, the 3rd year of the study was repeated to reveal the effect of grazing.
6.The figures do not have footnotes where the meaning of the letters is indicated and to what degree of significance they were compared.
----- The data given in the figure are based on averages. Since statistical analysis results were not given, statistical descriptive information in the form of degrees of freedom was not given under the figure.
- It is necessary for the authors to determine the total digestible nutrients and amounts of total and digestible energy of the groups of plants. They can do it with formulas.
-----Dear Editor, the features you have specified were not calculated in our study.
8.The authors conclude the use of a concentrate supplement for sheep. However, they do not make a deep discussion of the contribution of energy and consumption of plants. It is necessary to make an estimate of the requirements of the sheep.
--- We agree with the referee. Since the data evaluated in the study did not have the dry matter and energy values consumed by the animals, these statements in the manuscript are removed.
Reviewer 2 Report
Reviewed manuscript "Effects of Prickly Burnet (Sarcopoterium spinosum (L.) Spach.) Control and Sheep Grazing on Botanical Composition, Dry Matter Yield and Rangeland Quality on Gökçeada Island (Im-bros), Turkey" (animals-1942429) contains the results of interesting research work of scientific and practical significance.
The experiment was planned properly and carried out on sufficiently numerical material.
Statistical analysis of the obtained results is correct.
Tables presented the results and statistical data were constructed properly.
The discussion was carried out properly and the literature used in this part of the manuscript was chosen accordingly.
However, the manuscript contains several inaccuracies:
- the Simple Summary could be expanded, but the Abstract is too long and does not need to be split into Background, Methods, Results, Conclusions
- it is worth specifying what species of sheep was used for grazing
- line 44- correct the Latin name
line 51 - gra-zing - should be improved,
line 53 - spe-cies - should be improved
line 61 - pre-sent - should be improved
line 60 - bene-fit - should be improved
line 185-194 - font size should be corrected,
In general, manuscript should be adapted to the requirements of the journal.
In summary - the manuscript contains valuable results and after minor corrects should be publishing in Animals.
Author Response
Response to Reviewer-2
---Reviewed manuscript "Effects of Prickly Burnet (Sarcopoterium spinosum (L.) Spach.) Control and Sheep Grazing on Botanical Composition, Dry Matter Yield and Rangeland Quality on Gökçeada Island (Im-bros), Turkey" (animals-1942429) contains the results of interesting research work of scientific and practical significance.
---The experiment was planned properly and carried out on sufficiently numerical material.
---Statistical analysis of the obtained results is correct.
---Tables presented the results and statistical data were constructed properly.
---The discussion was carried out properly and the literature used in this part of the manuscript was chosen accordingly.
---However, the manuscript contains several inaccuracies:
---the Simple Summary could be expanded, but the Abstract is too long and does not need to be split into Background, Methods, Results, Conclusions
- it is worth specifying what species of sheep was used for grazing
--- Sheep (Gökçeada sheep) in the project carried out on the rangeland plots established in the upper part of the Yıldız Koyu (the name of the site of experiment), located in the north of Gökçeada (Imbroz) island, were selected from among a herd of 150 sheep roaming freely in rangeland, come to the corral when “called (with the sounds of the sheep owner)” and are accustomed to being fed, and were randomly distributed to plots depending on their age, live weight and condition. A total of 40 randomly selected sheep were between the age of 3-4 years and has an average live weight of 31.18 kg.
- line 44- correct the Latin name
---Corrections done.
line 51 - grazing - should be improved,
-- Corrections done.
line 53 - species - should be improved
-- Corrections done.
line 61 - pre-sent - should be improved
-- Corrections done.
line 60 - bene-fit - should be improved
-- Corrections done.
line 185-194 - font size should be corrected,
-- Corrections done.
In general, manuscript should be adapted to the requirements of the journal.
In summary - the manuscript contains valuable results and after minor corrects should be publishing in Animals.

Reviewer 3 Report
File attached

Author Response
Response to Reviewer-3
Summary
I would avoid subdividing into subparagraphs
Material and methods the description of this paragraph is somewhat confusing and incomplete in places. For example, you write that the botanical composition in the experimental areas was determined in May 2013, while in Appendix A you give the botanical composition for 2011, 2012 and 2013.
---Corrections done.
The paragraph should be revised to avoid mixing the description of operational aspects with those other aspects (e. g. animal management (line 80/84) with the physical description of the area) and should be edited in the following order: physical description of the site;
climatic data;
---- Corrections done.
vegetation;
---- Corrections done.
number and characteristics of the sites;
----- Corrections done.
animals (how many animals and in how many plots? Were the plots in the same area or in different locations?);
---- A total of 40 sheep were selected between the age of 3-4 years and they had an average live weight of 31.18 kg. Each grazing plot is placed on a field with 7.5 da and the protected plots were consisted of 1.5 da. Free grazing is done in the rangelands of Gökçeada (Imbroz). For this reason, the fencing was done around the plots, primarily to protect the planted plants from animals in the experimental field. The process of fencing has been done between 16-18.02.2011 by using a wicker iron cage with a length of 5 m and a height of 1.1 m. Paddocks and fences were established into the interior parts of the plots for grazing were placed just before grazing in order not to damage the natural form of rangeland. One side of the paddocks has left open so that the animals would graze freely in the rangeland. That is why, a total of eight paddocks were placed, one for each application, in order to meet the physiological needs of the sheep. The paddock area is planned to be 1.5 m2 for each sheep by taking into account the birth and lactation period of the animals (Altan et al., 2006). Accordingly, each paddock had an area of 7.5 m2.
study of botanical composition at the sites during the research period;
--- Distribution of species constituting the vegetation was determined in the experiment. Vegetation cover was measured using the ring (loop) method developed by Parker and Harris (1959) and recommended by Bakır (1970) for arid rangelands in our country, and used by many researchers (GökkuÅŸ, 1994; Şılbır and Polat, 1996; Koç and Çakal, 2004) in their researches. In each plot, 10 tracks were measured. The measuring tracks were fixed and the same tracks were measured throughout the experiment.
statistical analysis.
----
Please edit this chapter more clearly.
--- Line 2 Effects of Prickly Burnet (Sarcopoterium spinosum (L.) Spach.)……….
-- Effects of Prickly Burnet (Sarcopoterium spinosum (L.) Spach.) Control and Sheep Grazing on Hay Yield and Quality on Gökçeada (Imbros) Island, Turkiye
17 Sarcopoterium spinosum - please, type in italic all the nomenclature in the text
----- Corrections done.
21 About 0.15 ha of rangeland was allocated to each sheep and Five sheep were placed in each plot measuring 0.75 ha.
----- Corrections done.
32 Conclusions: (see above) However, Cutting is difficult over stony and rough terrain and pulling out creates erosion on sloping surfaces while burning……….. continue
----- Corrections done.
41 developed check the text, because these typos are common
----- Corrections done.
79-84 move the period in the part of M. and m. where you describe the management of the animals (see comments above)
------ Free grazing is done in the rangelands of Gökçeada (Imbroz). For this reason, the fencing was done around the plots, primarily to protect the planted plants from animals in the experimental field. The process of fencing has been done between 16-18.02.2011 by using a wicker iron cage with a length of 5 m and a height of 1.1 m. Paddocks and fences were established into the interior parts of the plots for grazing were placed just before grazing in order not to damage the natural form of rangeland. One side of the paddocks has left open so that the animals would graze freely in the rangeland. That is why, a total of eight paddocks were placed, one for each application, in order to meet the physiological needs of the sheep. The paddock area is planned to be 1.5 m2 for each sheep by taking into account the birth and lactation period of the animals (Altan et al., 2006). Accordingly, each paddock had an area of 7.5 m2.
104-105 ,.....supplementary and December 2013 and the sheep roamed at will over … not clear, please rephrase
------ Paddocks were placed in each plot to meet the water, supplementary and December 2013 and the sheep roamed at will over the rangeland throughout this period.
299 however, the ratio of herbaceous … write: however, the ratio of herbaceous species decreased with grazing, confirming studies in other Mediterranean environments where the presence of wild grazers predominates over livestock [* http://dx.doi.org/10.1080/03949370.2015.1022906, **https://doi.org/10.1007/s10344-019-1284-4 ***https://doi.org/10.3390/ani12060687].
--------- Corrections done.
Round 2
Reviewer 1 Report
Manuscript was improved
Reviewer 3 Report
Dear authors,
the suggestions have been taken into account to a large extent, mainly in favour of greater clarity of the experimental procedures. The results and discussions have also been better formulated for the benefit of the reader. As a final recommendation, especially for the rewritten parts, I suggest paying more attention to the English form.
Best regards